# The Development of iA CuffLink for Rotator Cuff Repair Telerehabilitation

**DOI:** 10.3390/s25206417

**Published:** 2025-10-17

**Authors:** Connor Luck, Rachel E. Roos, Jennifer Lambiase, Michelle Riffitts, Leslie Scholle, Simran Kulkarni, Dharma Parmanto, Vayu Putraadinatha, Made D. Yoga, Stephany N. Lang, Erica Tatko, Jim Grant, Jennifer I. Oakley, Ashley Disantis, Andi Saptono, Bambang Parmanto, Adam Popchak, Kevin M. Bell, Michael P. McClincy

**Affiliations:** 1Department of Orthopaedic Surgery, University of Pittsburgh, Pittsburgh, PA 15224, USA; oakleyji@upmc.edu (J.I.O.);; 2Department of Bioengineering, University of Pittsburgh, Pittsburgh, PA 15260, USAkmb7@pitt.edu (K.M.B.); 3ēlizur, Pittsburgh, PA 15237, USAstephanylang@elizur.com (S.N.L.);; 4Department of Health Information Management, University of Pittsburgh, Pittsburgh, PA 15260, USA; 5Department of Physical Therapy, University of Pittsburgh, Pittsburgh, PA 15219, USAajp64@pitt.edu (A.P.)

**Keywords:** rotator cuff repair, CuffLink, telerehabilitation, Delphi, questionnaire

## Abstract

**Highlights:**

**What are the main findings?**
A Delphi study was used to identify consensus in rehabilitation, informatic needs, and interface modalities for postoperative RCR patients using the iA CuffLink mHealth system.Participants in our pilot evaluation were generally pleased with the ease of use, information arrangement and usefulness of CuffLink as a telerehabilitation system.

**What is the implication of the main finding?**
These findings enhance our understanding of the informatics and usability needs in telerehabilitation systems and offer a solution for adopting technology in the early stages of postoperative RCR rehabilitation.

**Abstract:**

Proper rehabilitation following rotator cuff repair (RCR) is necessary for successful postoperative outcomes, though the average course of physical therapy (PT) is lengthy and costly. The goals of this study were to (1) develop exercise programs for the CuffLink mHealth system and (2) evaluate early prototype efforts at meeting the needs of RCR clients. A panel of 24 clinicians participated in a Delphi study to identify consensus in rehabilitation, key informatic needs, and appropriate interface modalities for client usage. Utilizing the Delphi findings, the iA CuffLink mHealth system was developed, and a pilot evaluation assessed the feasibility and usability of CuffLink through the mHealth App Usability Questionnaire (MAUQ). During the pilot evaluation, the overall MAUQ score was 6.14. All participants (n = 18) viewed messaging the care team and a real time rep counter as “important” or “very important”. All participants either agreed or strongly agreed that quantifying progress would help motivate them to be compliant, and that the app helped them achieve their recovery outcomes compared to the shoulder device alone. Participants were generally pleased with the ease of use, information arrangement, and usefulness of CuffLink. These findings can advance our understanding of the informatics and usability needs in telerehabilitation systems.

## 1. Introduction

Rotator cuff tears are the leading cause of shoulder injury in adult patients presenting with shoulder pain [1]. In the United States alone, rotator cuff tears generate over 4.5 million clinical visits, 250,000 surgeries, and $12 billion in healthcare costs annually [2,3]. Proper rehabilitation following rotator cuff repair (RCR) is critical for successful postoperative outcomes to restore native function [4]. The average course of rehabilitation requires up to 6 months of physical therapy (PT) sessions and generates costs between $2800 to $3400 [4]. Along with this, insurance protocols can cause delays in rehabilitation and impact the number of PT visits allotted to patients post-operatively [5,6].

Postoperatively, 81.2% of RCR patients initiate formal rehabilitation, with over 90% of this population utilizing PT services [7]. Failure to complete or inadequate post-operative PT treatment is associated with prolonged symptoms, a decline in clinical outcomes, a greater perceived number of barriers to exercise, and a lower overall quality of life [8,9,10]. The importance of patient compliance has been thoroughly investigated [8,11], and developing a method to promote compliance post-operatively could significantly improve clinical outcomes in the post-surgical population. Other diagnoses (knee osteoarthritis) have demonstrated that home-based therapy does not result in inferior outcomes compared to one-to-one or group-based therapy [12], suggesting promise for alternate and more cost-efficient forms of post-operative PT treatment. However, due to the precarious nature of the rotator cuff following surgical repair, as demonstrated by retear rates >50% in cohorts who experienced large tears (≥3 cm) [13,14], monitoring of exercise safety remains a concern during the early post-operative period. Creating a home-based therapy system that allows for monitoring of exercise performance and safety while not utilizing outpatient PT visits would address these challenges. Understanding the current gold standard RCR postoperative rehabilitation processes and identification of desirable features to include in a remote system for RCR rehabilitation is an essential step in developing a home-based exercise platform for shoulder rehabilitation.

The CuffLink system consists of a validated home-based shoulder exercise device (SSS-Strength and Stabilization System) instrumented with mechanical sensors integrated into a telerehabilitation system (iA-interACTION) which educates clients, provides real-time feedback, and allows remote monitoring by physical therapists. The goals of this study were to (1) utilize a Delphi process to define and develop post-RCR exercise programs for the CuffLink mHealth system, and (2) evaluate our early prototype efforts at meeting the informatics needs of the RCR clients and physical therapists following RCR.

## 2. Materials and Methods

### 2.1. Delphi Survey

Prior to development of the telerehabilitation system, an expert panel of twelve physical therapists and twelve orthopedic surgeons partook in a Delphi study to identify consensus in RCR rehabilitation and isolate key informatic needs and appropriate interface modalities for client usage [15]. Panelists were selected from geographically diverse locations throughout North America and had clinical expertise in shoulder injury rehabilitation, specifically for rotator cuff repairs. Consensus was defined a priori as ≥75% agreement. This threshold percentage is the standard for Delphi consensus, and thresholds above or below 75% have been shown to influence pattern agreements among reviewers following controlled feedback [16,17].

Surveys were distributed via email link for three rounds. Panelists were presented with seventeen free response questions about their experience with RCR. Common themes were identified as the “modal response” and themes reported by ≥25% of panelists were identified as the “second-tier response.”

### 2.2. CuffLink mHealth Development and Validation

Utilizing the findings from the Delphi study, the iA CuffLink mHealth system was developed using an iterative process of refining the app’s prototypes following the agile methodology. While the initial code base for the app was created prior to the Delphi study as a part of a different Telerehabilitation effort, in this project, the code base was adapted to become the app that we use for this study after the Delphi study. This allowed the content and the interactivity developed within the app to match with the need of the research. We used cross-platform technology to develop the app, specifically, ReactJS (Menlo Park, CA, USA) and NodeJS (Cologne, Germany), to build the technology stack for the app. The result of this development is an mHealth system, comprising two main front-end components: (1) a mobile app for the client, and (2) an online portal that is connected to the mobile app for the treating physical therapists. The mobile app is installed on the client’s smartphone and primarily functions as both a platform to deliver exercise related educational materials and exercise modules prior to any RCR exercise activity and an exercise tracking method during the RCR exercise activity. The mHealth app is connected to CuffLink via Bluetooth and provides real-time feedback for the client by pulling sensor data and synchronizing the data with the exercise modules. The CuffLink system uses Bluetooth Low Energy (BLE) with data arriving as base64-encoded values. The data is transmitted at 60 Hz with the app collecting the incoming data in memory and storing them to the private location of the app data files every second. Automated reconnection will be utilized when the Bluetooth connections are lost in which the data stream will be paused. Upon reconnection, data streaming resumes using the original baseline values and calibration factors to prevent missing data during session. The system uses BCrypt encryption for password storage and JWT tokens for user authentication with automatic refresh mechanisms. The synchronization of data from sensors with the exercise allows clients to follow the prescribed training accurately, ensuring that they complete their regime properly. The visualization of the data follows the recommendation by the expert panel and was inspected during development by our clinicians to be appropriate for delivering the information needed for the training sessions at home.

Sensor data successfully received are buffered and stored locally first before being transmitted to the server. This allows the app to work ‘offline’ and only push the data to the server when the connection can be established. The system does not need to stream the data to the server in real-time as all sessions are recorded locally first before being synchronized to the online portal post session. HTTPS encryption is used for all data transmissions. Role-based access control is used to manage the permissions for clients, clinicians, and virtual clinical specialists on the online portal. Through the online portal, treating physical therapists were able to remotely monitor their clients’ progress and configure exercises to personalize exercise modules to progress the rehabilitation toward their client’s specific goals. These configurations can be pushed back to the mobile app, allowing remote delivery of the exercise via Telerehabilitation. The flow of the system is illustrated in Figure 1.

A pilot evaluation assessed the feasibility and usability of the CuffLink mHealth system, utilizing the mHealth App Usability Questionnaire (MAUQ) that employed (1) a 7-Likert scale (1 = strongly disagree to 7 = strongly agree) [18] and (2) an exit survey [18,19]. All participants consented to participate in this IRB-approved study. Participants consisted of 18 postoperative RCR patients (age: 57.2 ± 7.4 yrs; time from surgery to study participation: 96.9 ± 45.4 days) across 3 performing providers. Thirteen of the participants underwent RCR on their right shoulder, and five underwent RCR on their left shoulder. Participants were verbally guided through the use of the app, and each completed a training course comprising eleven exercises that mimic a typical home session (Figure 2). Following this, the participants completed another session of eleven exercises at home with the same verbal guidance delivered via telerehabilitation. The participants completed the MAUQ after each session and completed the exit survey after the home session only. One participant did not complete their at-home visit due to an increase in pain and thus only completed surveys for their in-clinic test visit. Thirty-five responses were collected for the MAUQ, and seventeen participants responded to the exit survey.

## 3. Results

### 3.1. Delphi Survey

Three rounds of the Delphi survey were conducted before all questions reached consensus. All twenty-four panelists (100%) responded in the first round. In the second round, twenty-two panelists (92%) responded to all questions, and one panelist responded to the first twelve questions. In the third round, twenty-one panelists (88%) responded to the five remaining questions that had not reached consensus.

For post-operative ROM precautions, consensus was reached (96% agreement) that precautions are dependent on the repair and what procedure was performed, and that ROM is usually performed within a modified range starting with passive motion before progressing to active assisted/active ROM. Panelists agreed that ROM precautions should last for 3–8 weeks (95% agreement), with passive ROM starting 1–4 weeks post-operatively (96% agreement), active assisted ROM starting 4–8 weeks post-operatively (91% agreement), and active ROM starting 6–8 weeks post-operatively (87% agreement).

For glenohumeral motion, consensus was reached that the quantitative monitoring of flexion, internal/external rotation, and abduction would be desirable, and that passive and active ROM should be recorded (90% agreement). Panelists also agreed that measuring scapulothoracic rhythm would be important but not critical (81% agreement), and that the system should track compliance if possible (91% agreement).

For system usage, consensus was reached (91% agreement) that the patients should use the system prior to RCR surgery to familiarize themselves and address any pre-operative limitations or restrictions. All panelists agreed that they would have the patient use the system for 4–12 weeks following surgery or until symmetric motion is restored. Ninety-five percent of panelists would like the patient to use the system 1–3 times per week and increase this frequency if ROM is not progressing.

For the clinician interface, consensus was reached (96% agreement) that compliance and performance related metrics including frequency of use, ROM, and ROM progress should be displayed. All panelists agreed that they would like to see quantitative details about ROM displayed in a graphical format. For the patient interface, consensus was reached (91% agreement) that patients should see their ROM, ROM goals, and progress. A strong majority, 86% of panelists, agreed that patients should see this data in real time during exercise displayed on a chart or graph.

For remote system access, 95% of panelists reported that accessing the system from a web-based portal on a laptop or desktop computer could be easily used in the clinic and a mobile application on a tablet/smartphone could be easily used outside the clinic. Consensus for all Delphi questions can be seen in Table 1.

### 3.2. CuffLink mHealth Development and Validation

Among all participants, thirty-four responded to the MAUQ. The overall MAUQ score (1 = strongly disagree to 7 = strongly agree) was 6.14. The ease of use and satisfaction (MAUQ_E) was scored at 6.37, demonstrating that the participants generally perceived the app as easy to use. The system information arrangement (MAUQ_S) was scored at 6.01, demonstrating that the participants could generally find the information that they need. The usefulness (MAUQ_U) was scored at 6.00, demonstrating that the participants identified the usefulness of the app to achieve their goals. However, participants did log some complaints about the software during the evaluation, commenting that “there were obviously software problems or communication problems between the sensors and the app”, and that there were “some minor computer problems.”

Among all participants, seventeen completed the exit survey. All participants viewed messaging the care team as “important” or “very important”, and 94% of participants (16/17) viewed video calling the care team as “important” or “very important”. Fourteen out of seventeen (82%) of participants would like to virtually meet with their care team 1–3× a week, with two participants wanting to meet on-demand and one participant wanting to meet daily. All participants viewed a real time rep counter as “important” or “very important”, and 94% of participants (16/17) viewed a real time ROM display as “important” or “very important”. Eighty-eight percent of participants (15/17) wanted to have access to the system anywhere from 0–12 weeks following RCR, with two participants preferring to have access for however long their surgeon prescribes. All participants either agreed or strongly agreed that quantifying progress overtime would help motivate them to be compliant, and 94% of participants (16/17) either agreed or strongly agreed that collecting ROM data throughout a treatment protocol would improve patient-provider communication. All participants either agreed or strongly agreed that the iA app helped them achieve their recovery outcomes compared to the shoulder SSS device alone. When asked for additional input, there were some concerns, one of which noting that “the biggest problem was the sensors becoming disconnected”, and another stating that “[a] counter would be helpful.”

## 4. Discussion

Based on our findings from the Delphi Survey, clinicians agreed that patients should familiarize themselves with the system pre-operatively and utilize the system post-operatively for 4–12 weeks at a frequency of 1–3 times per week, with exercises progressing from passive to active ROM. Additionally, consensus was reached that the system should quantitatively monitor flexion, internal/external rotation, and abduction, movements for which its reliability and validity in tracking shoulder ROM have already been evaluated [20]. Clinicians agreed that patient compliance should be tracked, and performance related metrics should be displayed on the clinician interface with quantitative ROM details in a graphical format, accessible on either a web-based portal or mobile application. For patient interface, consensus was reached that ROM, ROM goals, and progress should be displayed in real time during exercise either by chart or graph. Following system development, clients generally agree that the iA CuffLink mHealth system app is easy to use, can generally find the information that they need, and can identify the usefulness of the app to achieve their rehabilitation goals. Responses from the exit surveys revealed that messaging/calling the care team 1–3 times per week and receiving real time exercise feedback is “important” or “very important” to patients. Additionally, patients would like to have access to the system for up to twelve weeks post-operatively, where their progress can be quantified and collected overtime to aid in compliance and patient-provider communication.

The balance between post-operative immobilization versus early range of motion following RCR surgery is well described, with early ROM risking disruption of the tendon repair and delayed ROM risking shoulder stiffness [21,22,23,24,25,26]. Our Delphi panelists corroborated this attitude, agreeing that post-operative passive ROM exercises are beneficial to combat stiffness, but they consistently highlighted the need for adaptive protocols to account for the variability in RCR surgeries (size of tear, quality of repair, etc.). To ensure proper healing, feedback and communication is imperative in the early phase of rehabilitation, a sentiment stressed by both the clinicians in the Delphi study and participants in the pilot evaluation study. The pilot evaluation participants felt that visual feedback is particularly helpful for tracking progress over time and during training in real time, as this aids motivation and ensures correct form during system use. The Delphi panelists similarly reached consensus that patients should be able to see their exercise data in real time displayed in a chart or graphical format. The iA CuffLink mHealth system app will provide a safe route for initiation of initial exercise protocol, optimizing recovery from stiffness while enabling modification through real time feedback and communication with the health care team.

Travel, costs, and time are examples of barriers that may contribute to failure in adherence to PT exercises, and socioeconomic status has been shown to impact the attendance of scheduled PT visits in-person post-operatively [27]. Patient compliance is critical for post-operative recovery [8,11], and failure to comply with PT exercises, which is prevalent in the RCR patient population [28], has been shown to result in low clinical outcomes, lingering symptoms, and lower quality of life [8,9,10]. In line with these findings, the Delphi panelists communicated the importance of patient compliance, agreeing that the system should track compliance if possible. Telerehabilitation was introduced to revolutionize the delivery of recovery services and has proven to provide effective rehabilitation compared to face-to-face therapy [12,29,30,31]. A recent study that evaluated the efficacy of telerehabilitation for patients with a RCR treated under workers’ compensation showed that patients who utilized telerehabilitation demonstrated earlier return to work compared to those who underwent only traditional PT [32], suggesting an increase in compliance in the telerehabilitation group. In support of these findings, participants using iA CuffLink were satisfied to conduct independent exercises at home.

During our pilot evaluation, those who had been familiarized with the system generally agreed that the mHealth app is easy to use. A participant stated that the app was “easy to use… I like the counter feature… it will be nice when it’s able to communicate with provider[s].” However, the ease of use can be improved by making the interface components configurable to the needs and limitations of the clients. For example, having the option to enlarge text size, or virtual buttons being paired with external buttons for pressing purposes, are some components that participants believe could improve ease of use and satisfaction for CuffLink. One complaint was that “the screen that I entered my data on was… smaller than it should be. Overall, the app was great otherwise.” Although participants could generally find the information that they needed, they felt that the information provided by the app could be improved by providing more detailed feedback on the exercise. This attitude was supported by our Delphi findings, in which the panelists agreed that performance metrics, such as quantitative details about ROM, ROM goals, and progress toward those goals, should be displayed on the patient interface. Additionally, participants were not convinced that the exercise data is being used in therapy or being communicated to their therapists. A recent study demonstrated that patient adherence is influenced by clinician involvement in the PROM findings [33]. In light of this, the attitude of the participants in this study highlights the need for an effective communication pathway via the CuffLink mHealth app to facilitate communication between app users and their clinician and improve patient confidence in their progress.

While we were pleased with the development of our system and its adherence to clinician attitudes towards the early phases of telerehabilitation, this study has some limitations. The Delphi survey, though conducted across clinicians with expertise in rotator cuff repairs and from diverse locations, lacked sensitivity analyses and thus verification for the robustness of our findings. However, we believe our study’s methodology led to results that adequately captured the perspectives of healthcare experts in the field of rehabilitation. Additionally, this study did not evaluate factors that are necessary to predict the success of our system, such as long-term patient compliance and functional outcomes. Future studies should investigate these variables to demonstrate the value behind a telerehabilitation system such as CuffLink.

## 5. Conclusions

Overall, participants were generally pleased with the ease of use, information arrangement, and usefulness of the iA CuffLink mHealth system as a telerehabilitation system that can be used at home. The insights gained from the Delphi survey and subsequent pilot study highlight the essential informatic features needed for transforming this exercise device into a successful rehabilitation system, including communication with providers and detailed feedback on exercise.

Our results can serve to equip health practitioners and developers with an understanding of the informatics and usability needs involved in such a system. By addressing these needs, CuffLink has the potential to deliver RCR postoperative care that is useful, easy to use, and provides the necessary information to both patient and provider in the early stages of rehabilitation. This ongoing development may contribute to the adoption of emerging technology in the early phases of telerehabilitation in postoperative patients.

## Figures and Tables

**Figure 1 sensors-25-06417-f001:**
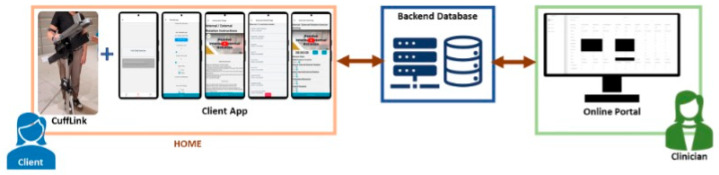
The flow of the CuffLink mHealth System, depicting sensor data from the client app being stored in a database prior to transmission to the clinician’s online portal. As suggested by the bidirectional arrows, data from the clinician’s online portal can be pushed back to the client app.

**Figure 2 sensors-25-06417-f002:**
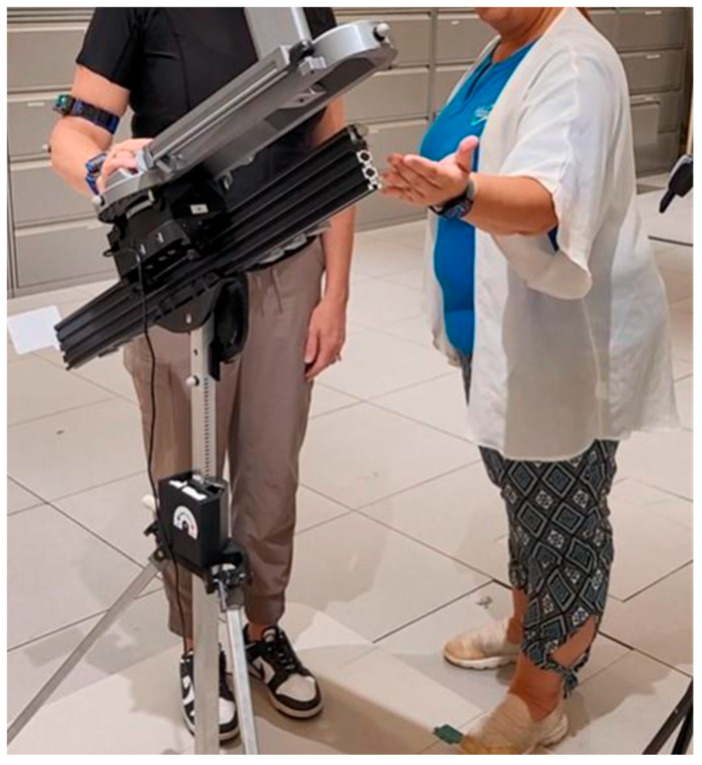
Postoperative RCR patient being verbally guided through the use of the iA app in concert with the SSS device by a clinical specialist in a controlled laboratory setting. Each participant completed 11 exercises consisting of both passive and active range-of-motion movements.

**Table 1 sensors-25-06417-t001:** Consensus reached among a panel of physical therapists and orthopedic surgeons to determine postoperative range of motion precautions, key informatic needs, and appropriate interface modalities for client usage of the CuffLink system.

	Delphi Survey Questions	Consensus (% Agreement)
1	Following a rotator cuff repair, what range of motion (ROM) precautions do you utilize?	Dependent on repair and procedure—start with passive ROM before progressing to active assisted/active ROM (96%)
2	For how long do you follow these ROM precautions?	3–8 weeks (95%)
3	Following a rotator cuff repair, when do you begin passive range of motion?	1–4 weeks post-operatively (96%)
4	Following a rotator cuff repair, when do you begin active assisted range of motion?	4–8 weeks post-operatively (91%)
5	Following a rotator cuff repair, when do you begin active range of motion?	6–8 weeks post-operatively (87%)
6	If you were to utilize a remote movement monitoring system for rehabilitation following a rotator cuff repair, what glenohumeral motions would you want to monitor?	Flexion, internal/external rotation, abduction (passive and active) (90%)
7	Would monitoring scapulothoracic rhythm be important to you?	Important but not critical (81%)
8	Would you want the device to track patient compliance?	System should track compliance if possible (91%)
9	Would you have the patient use the system before surgery?	Patient should use before surgery (91%)
10	If available to you, for how many weeks following surgery would you have a patient utilize a remote movement monitoring system?	4–12 weeks post-operatively (100%)
11	If available to you, how many times per week would you have your patient utilize a remote movement monitoring system?	At least 1–3 times per week (95%)
12	What information would you want the surgeon and/or physical therapist to see in a clinician-facing application?	Compliance and performance related metrics (frequency of use, ROM, ROM progress, etc.) (96%)
13	What kind of visualization do you think would be the most appropriate to deliver this information to the surgeon and/or physical therapist so they can make an informed decision about their patient/client?	Quantitative details about ROM displayed in a graphical format (100%)
14	What information would you want the patient to see in their patient-facing application?	ROM, ROM goals, and progress toward those goals (91%)
15	What kind of visualization do you think would be the most appropriate to deliver this information to the patient?	Data during exercise in real time displayed on a chart or graph (86%)
16	Do you prefer to access the information from the therapy session via a web-based portal accessed using a desktop computer or an app on a mobile platform (smartphone or tablet)? Or do you see yourself using both?	Use both the web-based portal and the mobile application (95%)
17	In what situation would these options be appropriate during your workflow?	Use web-based portal in the clinic and mobile application outside the clinic (95%)

## Data Availability

The generated datasets from this study are available upon request to the corresponding author.

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
