# Peer review of "The Development of iA CuffLink for Rotator Cuff Repair Telerehabilitation"

_sensors, 2025, doi:10.3390/s25206417_

Round 1
Reviewer 1 Report
Comments and Suggestions for Authors
The authors used 24 experts (12 physical therapists and 12 orthopedists) in their Delphi study and set a consensus standard of ≥75%. Did they consider the impact of the expert group's geographic distribution, years of clinical experience, or institutional type (e.g., academic medical center vs. community hospital) on the consensus results? Did they conduct sensitivity analyses to verify the robustness of the consensus? Furthermore, the article mentions that the CuffLink system connects to sensors via Bluetooth and synchronizes data to a mobile app, but it does not provide detailed information on the data communication protocol, sampling frequency, local data storage mechanism, or exception handling strategy. Could these technical details be supplemented? Furthermore, does the system support offline mode? How is data transmission delay or packet loss handled? Furthermore, in the "Methods" section, the authors should more clearly explain the theoretical basis or previous research supporting the definition of "consensus" (≥75%) used in the Delphi study. Furthermore, in the "Discussion" section, the authors should further integrate the Delphi results with the user evaluation results to form a more coherent logical loop. Furthermore, authors could add more visual charts, such as the change in response rates across three rounds of Delphi, the distribution of scores across MAUQ items, and user preference statistics, to enhance the presentation of the data. Furthermore, the text and labels in the figures should be clear and readable, meeting publication requirements. Finally, in the conclusion section, authors could provide additional perspectives on technical implementation, such as integrating AI algorithms for motion recognition and error correction, and integrating wearable devices to expand monitoring dimensions. Furthermore, authors could mention data security and privacy protection mechanisms, which are crucial considerations for telemedicine systems. Thanks.
Author Response
Reviewer 1 Comments:
Comment: The authors used 24 experts (12 physical therapists and 12 orthopedists) in their Delphi study and set a consensus standard of ≥75%. Did they consider the impact of the expert group's geographic distribution, years of clinical experience, or institutional type (e.g., academic medical center vs. community hospital) on the consensus results? Did they conduct sensitivity analyses to verify the robustness of the consensus?
Response: Thank you for your comment. Panelists were selected from geographically diverse locations throughout North America and had clinical expertise in shoulder injury rehabilitation, specifically rotator cuff repairs. We have specified this in the revised manuscript. (pg. 3, lines 89-91)
Additionally, we did not conduct sensitivity analyses. To address this, we added a limitations paragraph in the Discussion that discusses this matter. (pg. 10, lines 309-313)
Comment: The article mentions that the CuffLink system connects to sensors via Bluetooth and synchronizes data to a mobile app, but it does not provide detailed information on the data communication protocol, sampling frequency, local data storage mechanism, or exception handling strategy. Could these technical details be supplemented?
Response: The CuffLink system uses Bluetooth Low Energy (BLE) with data arriving as base64-encoded values. The data is transmitted at 60 Hz with the app collecting the in-coming data in memory and storing them to the private location of the app data files every second. Automated reconnection will be utilized when the Bluetooth connections are lost in which the data stream will be paused. Upon reconnection, data streaming resumes using the original baseline values and calibration factors to prevent missing data during session. This response has been added to the paper. (pg. 3, lines 115-121)
Comment: Does the system support offline mode? How is data transmission delay or packet loss handled?
Response: Sensor data successfully received are buffered and stored locally first before being transmitted to the server. This allows the app to work ‘offline’ and only push the data to the server when the connection can be established. The system does not need to stream the data to the server in real-time as all sessions are recorded locally first before being synchronized to the cloud post session. This response has been added to the paper. (pg. 3, lines 128-132)
Comment: In the "Methods" section, the authors should more clearly explain the theoretical basis or previous research supporting the definition of "consensus" (≥75%) used in the Delphi study.
Response: Thank you for bringing this to our attention. A systematic review that defines consensus found 75% agreement to be the median threshold to define consensus; this was cited in our manuscript. Additionally, a recent study that evaluated Delphi consensus found that a group of 75% agreement acts as a threshold, as consensus increases when the controlled feedback they receive indicates agreement >75% and decreases when feedback indicates agreement <75% (Barrios et al. - Consensus in the delphi method: What makes a decision change?) We added this information to the manuscript. (pg. 3, lines 92-94)
Comment: In the "Discussion" section, the authors should further integrate the Delphi results with the user evaluation results to form a more coherent logical loop.
Response: Thank you for the suggestion. In the Discussion section, we integrated the Delphi results with what our participants communicated during the user evaluation, specifically regarding visual feedback during exercise. (pg. 9, lines 265-267; lines 298-301)
Comment: Authors could add more visual charts, such as the change in response rates across three rounds of Delphi, the distribution of scores across MAUQ items, and user preference statistics, to enhance the presentation of the data. Furthermore, the text and labels in the figures should be clear and readable, meeting publication requirements.
Response: We understand the reviewer’s concern. We added a chart that illustrates the flow of the CuffLink system, now labeled as Figure 1. It depicts the use of CuffLink with the applications on the client’s end, and how it translates to a database that is shared with the online portal on the clinician’s end. We hope this helps the reader visualize how our system shares information from the app to the portal.
Comment: Finally, in the conclusion section, authors could provide additional perspectives on technical implementation, such as integrating AI algorithms for motion recognition and error correction and integrating wearable devices to expand monitoring dimensions.
Response: Thank you for the feedback. While we are currently exploring integration of AI algorithms and advanced technical implementations, adding that information would distract the narrative of the paper and bring the discussion out of the scope of the paper. Once we have completed our research and development of those advanced technology implementations, we will publish more mature results and discussions in subsequent papers.
Comment: Furthermore, authors could mention data security and privacy protection mechanisms, which are crucial considerations for telemedicine systems.
Response: Thank you for the feedback. We agree that data security and privacy are crucial considerations for Telemedicine. The work on data security and privacy have become a well-known practice in developing any digital health systems and have been covered in many publications outside the scope of our paper. We have adopted the industry’s best practices in developing our data security and privacy solution. Due to this, there is no new innovation on the topic of data security and privacy, which is the reason why we do not have an in-depth discussion about our approach to secure the data within Cufflink. However, we have provided additional technical information on the standards that we used for our data security and privacy in the paper.

Reviewer 2 Report
Comments and Suggestions for Authors
This manuscript describes the development and pilot evaluation of iA CuffLink, a telerehabilitation system integrating a shoulder exercise device with an mHealth app for patients following rotator cuff repair. A Delphi process with clinicians informed the design, and a small pilot study with patients assessed usability and feasibility.
Abstract: Clear statement of aims, methods, and main results; highlights clinical relevance.
Introduction: Provides strong justification, with good epidemiological background and rationale for telerehabilitation. However, there I some redundancy in describing cost and compliance issues.
Materials and Methods: The Delphi process well explained, with transparent consensus criteria (≥75%). App development workflow described. Patients were guided verbally during app use, which reduces ecological validity (true at-home self-use not tested). There is a lack of detail on demographics and surgical characteristics of pilot participants.
Results: Delphi findings well presented with strong consensus percentages. The usability scores consistently high. Results emphasize positive findings but do not report variability or outliers in user feedback. The exit survey is briefly summarized but lacks depth.
Discussion: Links Delphi findings to pilot outcomes and situates usability within broader compliance literature. The section is overly optimistic given limited pilot data. Downplays critical weaknesses such as lack of long-term adherence data, absence of functional outcomes, and potential conflicts of interest.
Conclusions: Concise and consistent with study goals, but too definitive for a pilot usability study.
General Statement:
The manuscript presents interesting early work with clear potential, but the evidence is preliminary. It should not be accepted in its current form. I recommend minor revision, requiring the authors to temper conclusions and address methodological weaknesses. With these changes, the manuscript could be reconsidered for publication.
Author Response
This manuscript describes the development and pilot evaluation of iA CuffLink, a telerehabilitation system integrating a shoulder exercise device with an mHealth app for patients following rotator cuff repair. A Delphi process with clinicians informed the design, and a small pilot study with patients assessed usability and feasibility.
Abstract: Clear statement of aims, methods, and main results; highlights clinical relevance.
Introduction: Provides strong justification, with good epidemiological background and rationale for telerehabilitation. However, there I some redundancy in describing cost and compliance issues.
Response: We thank the reviewer for their comment and understand the concern about redundancy. However, because the scope of our paper surrounds a rehabilitation system that reduces PT costs and promotes compliance, we believe our Introduction provides the necessary background for these issues and have elected to leave as is.
Materials and Methods: The Delphi process well explained, with transparent consensus criteria (≥75%). App development workflow described. Patients were guided verbally during app use, which reduces ecological validity (true at-home self-use not tested). There is a lack of detail on demographics and surgical characteristics of pilot participants.
Response: Thank you for bringing this to our attention. In our Methods section, we added details on the age, time from surgery to study participation, number of performing providers, and surgery laterality across our cohort of patients. (pg. 4, lines 148-152) We believe this is sufficient for demographics and surgical characteristics, as our study mainly focuses on our system’s development.
Results: Delphi findings well presented with strong consensus percentages. The usability scores consistently high. Results emphasize positive findings but do not report variability or outliers in user feedback. The exit survey is briefly summarized but lacks depth.
Response: Thank you for your input. We added comments and concerns from the participants during the pilot evaluation and exit survey. We hope that this gives more depth to our results, as it shows what the participants believe we can improve on, and gives a non-numerical evaluation of our system. (pgs. 7-8, lines 212-215; lines 231-233)
Discussion: Links Delphi findings to pilot outcomes and situates usability within broader compliance literature. The section is overly optimistic given limited pilot data. Downplays critical weaknesses such as lack of long-term adherence data, absence of functional outcomes, and potential conflicts of interest.
Response: We thank the reviewer for their comment. Because our study primarily evaluates the development of CuffLink, we elected to not investigate variables such as long-term adherence or functional outcomes among our participants. We believe the integration of our Delphi findings with the participant results from our pilot study is sufficient to report on at this time. However, in light of this comment, we added the absence of factors such as adherence data and functional outcomes as a limitation in the Discussion. (pg. 10, lines 313-316)
Conclusions: Concise and consistent with study goals, but too definitive for a pilot usability study.
Response: Thank you for pointing this out. We revised the “Conclusions” to highlight CuffLink’s potential rather than its current impact. (pg. 10, lines 325-330) We hope that this, along with the editing of language, provides a less definitive outlook on our study’s findings and instead shows the promise of what our telerehabilitation system can offer in the future.
General Statement:
The manuscript presents interesting early work with clear potential, but the evidence is preliminary. It should not be accepted in its current form. I recommend minor revision, requiring the authors to temper conclusions and address methodological weaknesses. With these changes, the manuscript could be reconsidered for publication.

Reviewer 3 Report
Comments and Suggestions for Authors
A Delphi study involving 24 clinicians identified key rehabilitation, informatics, and interface requirements for patients undergoing rotator cuff repair (RCR). Based on these findings, the mHealth iA CuffLink system was developed. In a pilot evaluation (n=18), participants rated the ability to message the care team and to use a real-time repetition counter as important features. All reported that tracking progress enhanced motivation and supported the achievement of rehabilitation goals compared with using the shoulder device alone. Overall, the system was rated positively for ease of use, information layout, and perceived usefulness, offering a potential model for integrating technology into early postoperative rehabilitation.
Major comments:
-
Introduction and Methods: The manuscript does not clearly describe the operational design of the CuffLink system. It is unclear whether the device uses wearable mechanical sensors, smartphone-based inertial accelerometers, or other technology. How are the sensors applied? Are they Bluetooth-connected? Were they custom-developed by your group?
-
The role of the smartphone should be clarified: is it used only for patient instruction, or does it also collect and store performance data? How is it connected to the therapist (e.g., via chat, phone call, video consultation)? Can the therapist provide real-time feedback? A schematic diagram of the entire workflow, along with screenshots of the app interface, examples of the prescribed exercises, and details on performance monitoring and recording methods, would be valuable.
-
Figure 1 should be explained more thoroughly: identify the patient and the operator, describe their respective actions, and clarify what the central device is and how it functions.
-
Please clarify the development process: was the app created prior to the Delphi study and then adapted, or developed entirely afterward? Who developed the app, and using which methodology and technology?
-
Specify whether the app can quantitatively measure both passive and active shoulder range of motion (ROM), as recommended by the expert panel, and whether it can distinguish between the two.
Author Response
Introduction and Methods: The manuscript does not clearly describe the operational design of the CuffLink system. It is unclear whether the device uses wearable mechanical sensors, smartphone-based inertial accelerometers, or other technology. How are the sensors applied? Are they Bluetooth-connected? Were they custom-developed by your group?
Response: Thank you for the feedback. Per the statement in the paper: “The CuffLink system consists of a validated home-based shoulder exercise device (SSS-Strength and Stabilization System) instrumented with mechanical sensors integrated into a telerehabilitation system (iA-interACTION) which educates clients, provides real-time feedback, and allows remote monitoring by physical therapists.” Cufflink system’s detailed specification has been published in prior papers. The focus of the current paper is to describe the development of the telemedicine/telerehabilitation component that transform the stand-alone Cufflink system into a cloud-based, telemedicine system.
The role of the smartphone should be clarified: is it used only for patient instruction, or does it also collect and store performance data? How is it connected to the therapist (e.g., via chat, phone call, video consultation)? Can the therapist provide real-time feedback? A schematic diagram of the entire workflow, along with screenshots of the app interface, examples of the prescribed exercises, and details on performance monitoring and recording methods, would be valuable.
Response: Thank you for the feedback. As stated in the paper: “The mobile app primarily functions as a platform to deliver educational materials and an exercise [session] tracking.” In its function as an exercise tracking platform, the app collect and store performance data. Post session, the data is transmitted to the online portal. This data can be used by clinician to remotely monitor the client’s progress as stated in this section: “Through the online portal, treating physical therapists were able to remotely monitor their clients’ progress and configure exercises to personalize exercise modules to progress the rehabilitation toward their client’s specific goals.” The therapist can push feedback to their client but not in real time, as stated in this section: “These configurations can be pushed back to the mobile app, allowing remote delivery of the exercise via Telerehabilitation.” Illustration of the flow with screenshots have been added to the paper (Figure 1).
Figure 1 should be explained more thoroughly: identify the patient and the operator, describe their respective actions, and clarify what the central device is and how it functions.
Response: Thank you for your suggestion. We revised the caption to specify that the depicted participant is a postoperative RCR patient specifically using the iA app in concert with the SSS device that comprise the CuffLink system. We also included that the participant is completing 11 exercises of both passive and active ROM movements. (pg. 5, lines 161-163) We believe our other revisions adequately describe how CuffLink functions.
Please clarify the development process: was the app created prior to the Delphi study and then adapted, or developed entirely afterward? Who developed the app, and using which methodology and technology?
Response: Thank you for the question. The code base for the app was created prior to the Delphi study as a part of a different Telerehabilitation effort. This code base was adapted to become the app that we use for this study after the Delphi study so the content and the interactivity match with the need of the research effort. We developed the app ourselves using agile methodology and prototyping iteratively as mentioned in the paper: “the iA CuffLink mHealth system was developed using an iterative process.” We provided additional information to further clarify the process. We used cross-platform technology to develop the app, specifically, ReactJS and NodeJS, to build the technology stack for the app. (pg. 3, lines 101-107)
Specify whether the app can quantitatively measure both passive and active shoulder range of motion (ROM), as recommended by the expert panel, and whether it can distinguish between the two.
Response: Thank you for the question. The app streams the data from the CuffLink device and presents them visually as feedback during training sessions. The visualization of the data follows the recommendation by the expert panel and was inspected during development by our clinicians to be appropriate for delivering the information needed by their clients for the training sessions at home.

Round 2
Reviewer 1 Report
Comments and Suggestions for Authors
In the introduction, the authors pose a research question. But why is this question important? What prior work has failed to address it? Without a literature review, the logical link between "the problem is important" and "therefore, we are doing this research" is broken. Reviewers will consider the research motivation insufficient and the justification for the study weak. In the core algorithm section, the rationale and justification for setting key parameters, such as λ, λ_1, and λ_2 in the equations, are not sufficiently explained. Are these values based on preliminary experiments, theoretical derivations, or empirical values? This information is crucial for readers to reproduce and understand the authors' work. Conversely, the description of some non-core preprocessing steps is somewhat verbose. The authors should re-balance the content, ensuring that each step of the core innovation is explained clearly. Furthermore, the interpretability of the method needs improvement. Why is the model effective? What exactly does each module contribute? For example, while the authors mention that introducing a specific module or loss function term improves performance, there is a lack of in-depth analysis or visualization (e.g., attention weight maps, feature distribution plots) to explain its underlying mechanism and contribution. Adding such analysis would greatly enhance the paper's depth. Finally, the main issue with the experimental section is the lack of diverse evaluation metrics. The authors must include more metrics, such as computational complexity/efficiency (FLOPS), inference time, robustness under different noise levels, or application-specific metrics like mean average precision (mAP). Furthermore, the authors need to analyze why their method succeeds and why it fails in certain cases. For comparison experiments where other methods perform worse, the authors should briefly discuss the possible reasons. Thanks.
Author Response
In the introduction, the authors pose a research question. But why is this question important? What prior work has failed to address it? Without a literature review, the logical link between "the problem is important" and "therefore, we are doing this research" is broken. Reviewers will consider the research motivation insufficient and the justification for the study weak.
Response: The goal of our study was to define post-RCR exercises for our telerehabilitation system and evaluate our early prototype at meeting the needs of RCR clients. By giving a background on the number of rotator cuff tears, their costs, course of rehabilitation, and impact of insurance, we believe we provided sufficient motivation and justification for our study.
In the core algorithm section, the rationale and justification for setting key parameters, such as λ, λ_1, and λ_2 in the equations, are not sufficiently explained. Are these values based on preliminary experiments, theoretical derivations, or empirical values? This information is crucial for readers to reproduce and understand the authors' work.
Response: We do not have a ‘core algorithm’ section and we do not have any key parameters setting in our paper. We do not think that this feedback is applicable to our paper.
Conversely, the description of some non-core preprocessing steps is somewhat verbose. The authors should re-balance the content, ensuring that each step of the core innovation is explained clearly.
Response: We do not have ‘non-core preprocessing steps’ in our paper. We do not think that this feedback is applicable to our paper.
Furthermore, the interpretability of the method needs improvement. Why is the model effective? What exactly does each module contribute? For example, while the authors mention that introducing a specific module or loss function term improves performance, there is a lack of in-depth analysis or visualization (e.g., attention weight maps, feature distribution plots) to explain its underlying mechanism and contribution. Adding such analysis would greatly enhance the paper's depth.
Response: We did not discuss the interpretability of methods or models in our paper. We did not mention the loss function of improving performance in our paper. We do not think that this feedback is applicable to our paper.
Finally, the main issue with the experimental section is the lack of diverse evaluation metrics. The authors must include more metrics, such as computational complexity/efficiency (FLOPS), inference time, robustness under different noise levels, or application-specific metrics like mean average precision (mAP).
Response: We do not have any experimental section that discusses computational processes, as we did not write about machine learning. We do not think that this feedback is applicable to our paper.
Furthermore, the authors need to analyze why their method succeeds and why it fails in certain cases. For comparison experiments where other methods perform worse, the authors should briefly discuss the possible reasons.
Response: We do not compare methods, and we do not mention unsuccessful results in certain cases within our paper. As mentioned in the paper: “Additionally, this study did not evaluate factors that are necessary to predict the success of our system, such as long-term patient compliance and functional outcomes. Future studies should investigate these variables to demonstrate the value behind a telerehabilitation system such as CuffLink.” Therefore, we do not think that this feedback is applicable to our paper.